# Aqua/Mechanochemical Mediated Synthesis of Novel Spiro [Indole–Pyrrolidine] Derivatives

**DOI:** 10.3390/ijms24032307

**Published:** 2023-01-24

**Authors:** Sodeeq Aderotimi Salami, Vincent J. Smith, Rui Werner Maçedo Krause

**Affiliations:** Department of Chemistry, Rhodes University, Grahamstown 6140, South Africa

**Keywords:** indole-2,3-diones, 3-dicyanomethylene-2H-indole-2-ones, isothiocyanates, spiro [indole–pyrrolidine, aqueous media, mechanochemistry

## Abstract

Spirocyclic scaffolds are found in many pharmacologically active natural and synthetic compounds. From time to time, efforts have been made to develop new or better processes for the synthesis of spirocyclic compounds. Spiro [Indole–pyrrolidine] Derivatives are readily synthesized in high to excellent yields by the Michael condensation of 3-dicyanomethylene-2H-indol-2-ones (produced via the Knoevenagel condensation of indole-2,3-dione with malononitrile) with isothiocyanate derivatives under aqueous and mechanochemical conditions. The advantages of this protocol are that the reactions are solvent-free, occur at ambient temperature, require short reaction times, have experimental simplicity, and produce excellent yields. These environmentally friendly reaction media are useful alternatives to volatile organic solvents.

## 1. Introduction

The spirocyclic motif can be found in chiral ligands, natural products, and molecules that are of pharmacological importance [1]. Spiro compounds have recently become more important in the development of pharmaceuticals due to the obvious conformational restriction imposed by the spiro atom, which allows the reduction of the entropy penalty associated with the binding to an active site of a protein target, a process that requires the adoption of a determined conformation [2,3]. Figure 1 shows some examples of spirocyclic natural products as well as spiro compounds of pharmaceutical interest, including acutumine, which is derived from the medicinal herb *Sinomenium acutum* and may have potential memory-improving properties, and vetivone, an antifungal agent produced by the fungus *Streptomyces griseus* [4,5].

Spirooxindoles have come to be recognized as privileged structures that can be found in both pharmaceutically relevant drugs as well as natural alkaloids including horsfiline, which was isolated from the Malaysian tree *Horsfieldia superba* [6,7,8]. Examples include potential candidates for targeted cancer therapy, antimalarial drugs, cell growth inhibitors such as spirotryprostatin A, fungicidal drugs such as welwitindolinone A, and compounds that may be employed in inhibitor-based treatments for tuberculosis (Figure 2) [9]. As a result, numerous synthetic techniques that target these molecules have been developed. Many of them make use of isothiocyanates/isocyanides as formal dipoles [10,11,12]. Due to the significance of these types of molecules, there has been interest in developing synthetic techniques for producing them, particularly in an enantioselective manner [13]. Cycloaddition reactions using exocyclic double-bonded cyclic molecules stand out from the rest due to their ease of operation and wide range of potential reaction partners [14].

Indoles are frequently found in nature and serve as the basic building block for several bioactive compounds. Numerous alkaloids and compounds derived from marine molluscs and shellfish, such as *rhynchophylline, elegantine, surogatoxin, neosurogatoxin,* etc., have recently been discovered to be heterocyclic compounds with a spiro system at the position 3 of the 2-indolinone skeleton. Many of these heterocycles are used as vital intermediates on the way to bioactive compounds [15,16,17]. Due to their numerous biological functions, derivatives of isothiocyanates are also quite interesting. The 3-spiroindolines incorporating pyrrolidine moiety appear to be potentially biologically active compounds considering the useful range of biological activities associated with naturally occurring spiroindole pyrrolidine [18].

Water is one of the most intriguing alternative response media because of its unique features. Many biochemical processes take place in aqueous media, which is the most abundant accessible molecule on the earth [19,20]. Because it induces the disintegration of organometallic reagents, which are only utilized in dry organic solvents, it has not been a preferred reaction media for organic chemists [21,22]. In reality, water is commonly employed to work up organic reactions and it has been connected with a waste-production stage, as well as the apparent challenges of cleaning up water from reactant residues [20]. However, in recent years, the image has changed, and aqueous medium has piqued the curiosity of organic chemists and others, since remarkable discoveries have been achieved using it [23]. On the other hand, mechanochemical methods have been used in synthesis for a long time, but their perception in the synthetic world has altered recently, and they are on their way to becoming mainstream [24,25,26]. However, the combination of mechanochemical synthesis with techniques meant to improve synthetic efficiency by enabling the generation of multiple bonds in a single operation is a relatively new approach to sustainable chemistry that has already shown to be very promising [27,28,29].

Aqua and mechanochemistry are commonly used for rapid organic synthesis as they are faster, more efficient, and simpler to employ than traditional techniques. In this regard, the synthesis of spiroindole pyrrolidine derivatives have been developed under mechanochemical and aqua conditions, and they are based on the use of water (aqueous reaction) and high-speed vibration milling with a double agate ball (6 mm diameter) in an agate jar and piperidine as a liquid-assisted grinding agent.

## 2. Results and Discussion

In continuation of our earlier interest in the synthesis of biodynamic heterocycles under aqua and mechanochemical conditions, we investigated the reaction of indole-2,3-dione (**1a**) with malononitrile (**2**) to give 3-dicyanomethylene-2H-indol-2- ones (**3a**) that was cyclocondensed in situ with 4-methoxyphenyl isothiocyanates (**4a**) under aqua and mechanochemical conditions leading to the facile one-pot synthesis of 2′′-oxo-5′-sulfanylidene-1′′,2′′-dihydrodispiro[cyclohexane-1,2′-pyrrolidine-4′,3′′-indole]-2,5-diene-3′,3′-dicarbonitrile (**5a**) (Figure 1).

The formation of (**5a**) was assumed to proceed via the Michael condensation, i.e., involving the attack of the carbanion nucleophile on the olefinic carbon of ylidene, followed by the cycloaddition of isothiocyanates on the cyano group to form the spiro compound.

The initial investigation began with the reaction between isatin (**1a**), malononitrile (**2**), and 4-methoxy isothiocyanate (**4a**) in methanol at room temperature for 6 h in the presence of piperidine as catalyst. Interestingly, a new product (**5a**) was observed and isolated in 70% yield (Table 1 entry 1). We investigated the impact of various solvents on the reaction course to determine the most suitable reaction conditions and to compare results of the aqua conditions with those obtained under standard conditions (organic solvents). (Table 1). The reaction proceeded smoothly in dichloromethane (CH_2_Cl_2_) and ethanol with high yields of 62% and 75% respectively (Table 1 entries 3, 4). Solvents such as toluene and diethyl ether were ineffective even after 6 h, producing trace amounts of the products (Table 1 entries 2, 5). When the reaction was carried out in water in the presence of two drops of piperidine, the product (**5a**) was obtained in high yield (**85%**) within 15 min (Table 1 entry 6). Additionally, when the reaction was carried out in the absence of catalyst (piperidine), the reaction failed to furnish the final spiro adduct, and instead furnished the intermediate (**3a**). This implies that the cycloaddition of isothiocyanates (**4a**) with 3-dicyanomethylene-2H-indol-2- ones (**3a**) requires a base for activation.

We established that the optimal conditions for the one-pot synthesis of spiro pyrrolidine are room temperature as well as the presence of piperidine as the catalyst and water as the reaction solvent. The NMR data of the synthesized spiro compound (**a**–**h**) clearly show the existence of a mixture of diastereomers. The results clearly indicate that all ^1^H NMR and ^13^C NMR peaks of compound (**5a**–**h**) are duplicated, revealing that the products (**5a**–**k**) exist as a mixture of diastereomers. The reaction carried out for isatin/5-bromo isatin (**1a, 1b**) and isothiocyanates derivatives (**4a**–**k**), bearing electron-donating **(4)** as well as electron-withdrawing groups (**4b, 4c, 4g, 4i, 4j, 4k**) furnished the corresponding 2′′-oxo-5′-sulfanylidene-1′′,2′′-dihydrodispiro[cyclohexane-1,2′-pyrrolidine-4′,3′′-indole]-2,5-diene-3′,3′-dicarbonitrile derivatives and 5′′-bromo-2′′-oxo-5′-sulfanylidene-1′′,2′′-dihydrodispiro[cyclohexane-1,2′-pyrrolidine-4′,3″-indole]-2,5-diene-3′,3′-dicarbonitrile derivatives in excellent yields (Figure 3).

### Mechanochemical Synthesis

In this study, we developed our system utilizing the flexible methodology laid out by Dandia et al. [13]. However, we found that high-energy ball milling was necessary to obtain satisfactory yields. In the initial experiment, a moderate yield (64%) was obtained by simply milling the two solids (isatin (**1a**)), malononitrile (**2a**) under neat conditions without the use of an auxiliary grinding material (Figure 2, Table 2 entry 1).

Pleasingly, treatment of isatin (**1a**) with one equivalent of malononitrile (**2**) in the presence of 2–3 drops of MeOH as additive afforded the desired 3-dicyanomethylene-2H-indol-2- ones (**3a**) in 69% yield after milling for 10 min (Table 2 entry 2). The yield increased further to 74% when DCM was employed as LAG (Table 2 entry 3). To further boost the yield, the reaction was conducted in the presence of water as the LAG agent, and the yield of (**3a**) rose rapidly to 90% (Table 2 entry 4), while extending the milling times to 15 min further increased the yield to 98% (Table 2 entry 5). A grinding auxiliary could serve a variety of purposes, such as enhancing mixing and facilitating energy transfer. Early studies into the cyclocondensation of the isothiocyanates focused on determining the ideal reaction time for the isolated step rather than two-step, i.e., the 3-dicyanomethylene-2H-indol-2- ones (**3a**) was isolated from step one and purified before being subjected to this second reaction optimization (Figure 3). Having achieved optimal conditions for the first step of the reaction, our attention turned to the second step.

For the second step of the reaction, we started with cyclocondensation of isothiocyanates (**4a**) with intermediate (**3a**), which furnishes the corresponding spiroindole–pyrrolidine in 83% yield after milling for 10 min in the presence of piperidine as additive. Increasing the milling time had a positive effect on the reaction yield, since the yield increases further to 97% at the milling time of 20 min. When the reaction was carried out in the absence of catalyst (piperidine), the reaction did not afford the anticipated product even after 1 h milling time.

The scope of the one-pot mechanochemical process was investigated following the established optimized reaction conditions (Figure 4). After 15 min of milling, the combined two-step procedure was unable to furnish product **5a** as anticipated. The reaction’s conditions remained unaffected by extending the milling time to an hour. As the reaction progressed, intermediate 3-dicyanomethylene-2H-indol-2-ones (**3a**) was obtained rather than the desired final spiro adducts (**5a**). This is due to electron donating resonance effect of indole of nitrogen, which decreases the electrophilicity of isothiocyanates carbon, thereby preventing the final cyclocondensation reaction to furnish compound **5a**.

The synthesis of 3-dicyanomethylene-2H-indol-2-ones (**3a**) according to the first step is experimentally simple, leading to the formation of analytically pure compounds in 98% yield. The structures were confirmed by ^1^ H and ^13^C NMR, IR spectroscopy, and mass spectrometry. We demonstrated that the cyclocondensation of isothiocyanates (**4a**–**h**) with 3-dicyanomethylene-2H-indol-2-ones (**3a**) in the presence of piperidine as catalyst afforded spiro compounds in excellent yield after 15 min milling at room temperature. 

The Knoevenagel condensation of 5-bromo, isatin (**1b**) and malononitrile (**2**) afforded the corresponding 5-bromo, 3-dicyanomethylene-2H-indol-2-ones (**3b**). Subsequent cyclo condensation with isothiocyanates (**4i**–**l**) furnishes the anticipated spiro-pyrrolidine derivatives (**5i**–**l**) in excellent yield after 30 min of milling (Figure 4 and Figure 5).

With the optimized reaction conditions in hand, we then examined the scope and limitations of the reaction. First, we examined the variation of the intermediate moiety. The parent 3-dicyanomethylene-2H-indol-2-ones (**2**) underwent a smooth reaction with different functionalized isothiocyanates, and the corresponding spiroindole–pyrrolidine derivatives were formed with excellent yield, (**3a**–**h**). Moreover, the substitution at the 5-position for 5-bromo 3-dicyanomethylene-2H-indol-2-ones (**3b**) did not affect the reaction outcome, and the desired product was formed in good to excellent yield and selectivity (**5i**–**l**). Next, we focused on the scope of the isothiocyanate component. The isothiocyanate with electron-donating and electron withdrawing groups of *N*-aryl ring (**4a**–**h**) were well tolerated, and the desired product was formed in good yield with moderate diastereoselectivity (**5a**–**h**). Additionally, cyclocondensation of 5-bromo, 3-dicyanomethylene-2H-indol-2-ones with 2-iodophenylisothiocyanate afforded the corresponding product in moderate yield with a mixture of diastereomers (**5i**). Additionally, the 2″,9,10-trioxo-5′-sulfanylidene-1″,2″,9,10- tetrahydro-4aH-dispiro[anthracene-2,2′-pyrrolidine-4′,3″-indole]-3″-dicarbonitrile synthesized from 2-isothiocyanatoanthracene-9,10-dione (**4h**) and 3-dicyanomethylene-2H-indol-2-ones (**3a**) underwent a smooth reaction leading to the formation of the desired products in high yield (**5h**).

## 3. Materials and Methods

Mechanochemical experiments were carried out in an in-house modified Makita (Jigsaw) ball mill that was equipped with reactors (diameter: 2.0 cm; height: 2.0 cm; volume of reactor: 13.2 mL). All reactors are stainless steel and equipped with stainless steel balls (diameter: 6 mm; mass: 0.90 g;). A FT-IR-Eco ATR Bruker Alpha II Spectrometer was used for the FT-IR analysis. The IR spectra were obtained by the attenuated total reflection (ATR) method. For each experiment, 16 scans were performed in the frequency range from 650 to 4000 cm^−1^. Melting points of all the compounds were determined using a Koffler hot-stage apparatus and are uncorrected. NMR spectra were recorded on a Bruker Advance III 400 spectrometer using CDCl_3_ or DMSO-d_6_ as a solvent with tetramethyl silane used as internal standard. LC-MS/MS data were recorded on a Bruker Compact quadrupole time-of-flight (QToF) mass spectrometer. Raw mass spectrometry data were processed using MZmine software (version 2.38). Solvents and chemicals used were of analytical grade, which were purchased from Sigma Aldrich and used without further purification. The purity determination of the starting materials and reaction monitoring was performed by thin-layer chromatography (TLC) on Merck silica gel G F254plates.

### 3.1. General Experimental Procedures for the Aqua Synthesis of Spiro [Indole–Pyrrolidine] Adducts in the Presence of Piperidine

A mixture of isatin/5-bromo isatin (**1a**/**1b**, 1 mmol) malononitrile (2, 1 mmol), and 4-methoxyphenyl isothiocyanate (**4a**, 1 mmol) was vigorously stirred in 2 mL water at room temperature for 15 min in the presence of 2–3 drops of piperidine. Upon completion, the organic layer was separated, and the combined organic phases were concentrated under reduced pressure and the residue was purified by column chromatography using DCM/Hexane (3: 1) as the eluent to afford the desired products. Typical yields range from 80 to 95%. All other products (**5a**–**l**) were obtained by a similar approach.

### 3.2. General Procedure for the Mechanochemical Synthesis of Spiro [Indole–Pyrrolidine] Adducts (3a/3b)

**Experiment 1.** A mixture of isatin/5-bromo isatin (**1a**/**1b**, 1 mmol), malononitrile (2, 1 mmol), and additive (water 50 µL) was milled in a 13.2 mL stainless steel milling vessel containing two balls of the same material (diameter: 6 mm; mass: 0.90 g) at 25 Hz for 20 min. After the milling was complete, the content in the milling vessel was transferred into a beaker using a small amount of organic solvent (DCM). Then, purification was carried out by column chromatography to afford the corresponding Passerini adducts in high to excellent yield.

### 3.3. General Procedure for the Mechanochemical Synthesis of Spiro [Indole–Pyrrolidine] Adducts (5a–l)

**Experiment 2.** A mixture of 3-dicyanomethylene-2H-indol-2-ones/5-bromo, 3-dicyanomethylene-2H-indol-2-ones (**3a**/**3b**, 1 mmol) and 4-methoxyphenyl isothiocyanate (**4a**, 1 mmol) with 2–3 drops of piperidine was milled in a 13.2 mL stainless steel milling vessel containing two balls of the same material (diameter: 6 mm; mass: 0.90 g) at 25 Hz for 20 min. After the milling was complete, the content in the milling vessel was transferred into a beaker using a small amount of organic solvent (DCM). Then, purification was carried out by column chromatography to afford the corresponding oxindole derivatives in high yield.


**4-methoxy-2″-oxo-5′-sulfanylidene-1″,2″-dihydrodispiro[cyclohexane-1,2′-pyrrolidine-4′,3″ indole]-2,5-diene-3′,3′-dicarbonitrile (5a).**


Brown solid (Yield: Met A 85%, Met B, 97%); m.p 142-144 °C. IR (NaCl) *v*(cm^−1^) 3324 (NH), 2220 (CN), and 1710 (CO). ^1^H NMR (400 MHz, DMSO) δ 11.22 (s, 1H), 9.06 (s, 1H), 7.89 (d, J = 7.8 Hz, 1H), 7.58 (t, J = 7.8 Hz, 1H), 7.14 (t, J = 7.9 Hz, 1H), 6.94 (d, J = 7.9 Hz, 1H), 6.85 (d, J = 8.6 Hz, 1H), 3.96−3.77 (m, 1H), 3.74 (s, 1H), 1.59 (d, J = 33.4 Hz, 2H), 1.23 (s, 4H)^. 13^C NMR (101 MHz, DMSO) δ 181.5, 164.6, 156.9, 151.0, 147.1, 138.0, 134.6, 127.8, 126.1, 123.7, 119.1, 113.2, 112.1, 81.0, 61.5, 55.5, 49.2, 29.6, 25.7, 22.2. HR-MS (ESI): [M+H^+^] calcd for C_19_H_14_N_4_O_2_S: 362.0837 found: 362.0998.


**2,4,6-trimethyl-2″-oxo-5′-sulfanylidene-1″,2″-dihydrodispiro[cyclohexane-1,2′-pyrrolidine-4′,3″-indole]-2,5-diene-3′,3′-dicarbonitrile (5b).**


Brown solid (Yield: Met A 93%, Met B, 92%); m.p 131–132 °C. IR (NaCl) *v*(cm^−1^) 3336 (NH), 2225 (CN), and 1720 (CO). ^1^H NMR (400 MHz, DMSO) δ 11.22 (s, 1H), 8.68 (s, 1H), 7.87 (d, J = 7.8 Hz, 1H), 7.57 (t, J = 7.7 Hz, 1H), 7.13 (t, J = 7.7 Hz, 1H), 6.93 (d, J = 8.0 Hz, 1H), 6.85 (s, 1H), 3.88 (s, 2H), 2.23 (s, 2H), 2.08 (s, 3H), 1.63 (t, *J* = 13.8 Hz, 2H), 1.54 (s, 2H).^.13^C NMR (101 MHz, DMSO) δ 180.7, 164.2, 151.0, 146.6, 138.0, 136.5, 128.5, 126.2, 123.3, 118.8, 113.3, 112.2, 81.2, 49.2, 25.8, 24.4, 20.9. HR-MS (ESI): [M+H^+^] calcd for C_21_H_18_N_4_OS: 374.1201 found: 374.1765.


**3,5-dimethyl-2″-oxo-5′-sulfanylidene-1″,2″-dihydrodispiro[cyclohexane-1,2′-pyrrolidine-4′,3″-indole]-2,5-diene-3′,3′-dicarbonitrile (5c).**


Brown solid (Yield: Met A 95%, Met B, 87%); m.p 140–142 °C. IR (NaCl) *v*(cm^−1^) 3362 (NH), 2218 (CN), and 1708 (CO). 1H NMR (400 MHz, DMSO) δ 11.23 (s, 1H), 10.56 (s, 1H), 10.47 (s, 1H), 10.03 (s, 1H), 9.59 (s, 1H), 8.22 (s, 1H), 7.87 (s, 1H), 7.68 (d, J = 7.7 Hz, 1H), 7.57 (d, J = 6.8 Hz, 1H), 7.23 – 7.15 (m, 1H), 7.12 (dd, J = 15.7, 7.9 Hz, 1H), 7.05 (d, J = 7.5 Hz, 1H), 7.04−6.96 (m, 1H), 6.94 (d, J = 6.8 Hz, 1H), 6.87−6.79 (m, 1H), 6.76 (s, 1H), 6.70 (s, 1H), 6.59 (s, 1H), 5.76 (s, 1H), 3.56 (d, J = 5.1 Hz, 1H), 2.23 (d, J = 6.0 Hz, 3H), 1.65 (t, J = 18.4 Hz, 3H). ^13^C NMR (101 MHz, DMSO) δ 184.9, 179.7, 167.8, 165.0, 159.7, 151.3, 140.1, 139.6, 139.4, 139.3, 138.3, 137.9, 128.6, 126.6, 125.5, 123.7, 121.8, 121.4, 120.8, 119.5, 117.2, 116.2, 115.6, 114.3, 114.0, 113.6, 109.7, 109.5, 106.4, 56.5, 53.7, 52.4, 44.2, 26.5, 26.2, 23.5, 21.5. HR-MS (ESI): [M+H^+^] calcd for C_20_H_16_N_4_OS: 360.1044 found: 360.3296.


**3-methyl-2″-oxo-5′-sulfanylidene-1″,2′′-dihydrodispiro[cyclohexane-1,2′-pyrrolidine-4′,3″-indole]-2,5-diene-3′,3′-dicarbonitrile (5d).**


Brown solid (Yield: Met A 91%, Met B, 95%); m.p 138–140 °C. IR (NaCl) *v*(cm^−1^) 3304 (NH), 2217 (CN), and 1710 (CO). ^1^H NMR (400 MHz, Acetone) δ 10.02 (s, 1H), 9.76 (s, 1H), 7.90 (d, J = 7.9 Hz, 1H), 7.48 (t, J = 7.8 Hz, 2H), 7.27 (d, J = 8.3 Hz, 1H), 7.16 (s, 1H), 7.07 (d, J = 7.8 Hz, 1H), 7.03 (d, J = 8.2 Hz, 2H), 6.93 (d, J = 8.0 Hz, 1H), 5.49 (s, 1H), 4.40 (d, J = 33.9 Hz, 2H), 1.92 (s, 3H). ^13^C NMR (101 MHz, DMSO) δ 188.1, 171.3, 164.3, 151.3, 147.1, 138.0, 129.6, 126.1, 123.3, 122.3, 119.1, 118.4, 113.5, 111.8, 81.0, 75.4, 72.9, 67.4, 64.9, 21.2, 17.3, 14.5. HR-MS (ESI): [M+H^+^] calcd for C_19_H_14_N_4_OS: 347.0888 found: 347.1028.


**2-oxo-5**
**′**
**-sulfanylidene-1,2-dihydro-4**
**″**
**aH-dispiro[indole-3,4**
**′**
**-pyrrolidine-2**
**′**
**,2**
**″**
**-naphthalene]-3**
**′**
**,3**
**′**
**-dicarbonitrile (5e).**


Brown solid (Yield: Met A 87%, Met B, 90%); m.p 156–158 °C. IR (NaCl) *v*(cm^−1^) 3355 (NH), 2220 (CN), and 1715 (CO). ^1^H NMR (400 MHz, DMSO) δ 11.21 (s, 1H), 11.05 (s, 1H), 10.53 (s, 1H), 10.44 (s, 1H), 7.86 (d, J = 7.8 Hz, 1H), 7.66 (d, J = 7.8 Hz, 1H), 7.56 (t, J = 7.7 Hz, 1H), 7.49 (d, J = 7.6 Hz, 1H), 7.21−7.07 (m, 2H), 7.06 (t, J = 8.6 Hz, 1H), 6.94 (s, 1H), 6.92 (t, J = 8.4 Hz, 2H), 6.83 (d, J = 7.7 Hz, 1H), 1.66 (s, 3H). ^13^C NMR (101 MHz, DMSO) δ 173.7, 167.8, 164.9, 164.1, 159.8, 152.5, 151.0, 147.0, 140.0, 139.3, 138.8, 138.2, 128.6, 126.6, 126.1, 125.1, 123.3, 121.3, 120.8, 119.4, 119.0, 114.2, 112.1, 109.7, 109.6, 81.0, 53.7, 52.4, 48.0, 23.2. HR-MS (ESI): [M+H^+^] calcd for C_22_H_14_N_4_OS: 382.0888 found: 382.3186.


**{3′,3′-dicyano-2″-oxo-5′-sulfanylidene-1″,2″-dihydrodispiro[cyclohexane-1,2′-pyrrolidine-4′,3″-indole]-2,5-dien-4-yl}azinic acid (5f).**


Brown solid (Yield: Met A 88%, Met B, 93%); m.p 153–155 °C. IR (NaCl) *v*(cm^−1^) 3328 (NH), 2212 (CN), and 1730 (CO). ^1^H NMR (400 MHz, DMSO) δ 11.44 (s, 1H), 11.24 (s, 1H), 11.17 (s, 1H), 11.06 (s, 1H), 10.58 (s, 1H), 10.48 (s, 1H), 7.89 (d, J = 7.8 Hz, 1H), 7.68 (d, J = 7.8 Hz, 1H), 7.63−7.53 (m, 2H), 7.54 – 7.45 (m, 1H), 7.42 (dd, J = 14.6, 6.6 Hz, 1H), 7.25 (d, J = 10.9 Hz, 1H), 7.17 (dd, J = 17.1, 7.8 Hz, 3H), 7.08 (ddd, J = 18.4, 11.9, 4.5 Hz, 3H), 7.01 (dd, J = 14.7, 7.1 Hz, 2H), 6.93 (dd, J = 14.8, 7.2 Hz, 3H), 6.83 (d, J = 7.7 Hz, 3H), 6.17 (s, 1H), 4.03 (dd, J = 14.2, 7.1 Hz, 1H), 3.81 (d, J = 5.0 Hz, 2H), 3.56 (d, J = 5.3 Hz, 5H), 2.73−2.56 (m, 2H), 2.09 (s, 21H), 1.68 (d, J = 8.0 Hz, 3H). ^13^C NMR (101 MHz, DMSO) δ 207.0, 184.8, 169.7, 167.8, 165.0, 164.2, 159.8, 151.2, 146.9, 140.1, 139.3, 138.8, 138.2, 131.9, 128.7, 126.6, 125.1, 123.6, 123.3, 122.9, 121.4, 120.8, 118.3, 116.3, 115.3, 114.3, 112.6, 109.7, 106.3, 81.0, 65.6, 60.3, 53.8, 48.1, 31.1, 23.5. HR-MS (ESI): [M+H^+^] calcd for C_18_H_12_N_5_O_3_S: 379.0739 found: 379.1835.


**2-iodo-2**
**″**
**-oxo-5**
**′**
**-sulfanylidene-1**
**″**
**,2**
**″**
**-dihydrodispiro[cyclohexane-1,2**
**′**
**-pyrrolidine-4**
**′**
**,3**
**″**
**-indole]-2,5-diene-3**
**′**
**,3**
**′**
**-dicarbonitrile (5g).**


Brown solid (Yield: Met A 91%, Met B, 96%); m.p 130–132 °C. IR (NaCl) *v*(cm^−1^) 3337 (NH), 2220 (CN), and 1718 (CO). ^1^H NMR (400 MHz, CDCl_3_) δ 10.23 (s, 1H), 10.07 (s, 1H), 9.56 (s, 1H), 9.47 (s, 1H), 6.86 (d, J = 7.8 Hz, 1H), 6.57 (t, J = 7.9 Hz, 1H), 6.50 (d, J = 7.4 Hz, 1H), 6.12 (dd, J = 14.5, 6.7 Hz, 2H), 6.08 (d, J = 8.1 Hz, 1H), 5.93 (t, J = 7.5 Hz, 2H), 5.84 (d, J = 7.6 Hz, 1H), 0.69 (d, J = 7.4 Hz, 2H). ^13^C NMR (101 MHz, CDCl_3_) δ 189.6, 172.7, 168.9, 164.3, 155.7, 151.4, 143.1, 130.9, 129.9, 128.0, 126.1, 123.7, 118.3, 116.7, 114.3, 85.8, 58.5, 57.2, 31.4, 28.2. HR-MS (ESI): [M+H^+^] calcd for C_18_H_11_IN_4_OS: 457.9698 found: 457.1710.


**2″,9,10-trioxo-5′-sulfanylidene-1″,2″,9,10-tetrahydro-4aH-dispiro[anthracene-2,2′-pyrrolidine-**
**4**
**′**
**,3**
**″**
**-indole]-3**
**′**
**,3**
**′**
**-dicarbonitrile (5h).**


Brown solid (Yield: Met A 76%, Met B, 84%); m.p 157–159 °C. IR (NaCl) *v*(cm^−1^) 3346 (NH), 2230 (CN), and 1715 (CO). ^1^H NMR (400 MHz, DMSO) δ 11.22 (s, 1H), 11.01 (s, 1H), 10.51 (d, J = 38.1 Hz, 1H), 7.88 (d, J = 7.8 Hz, 1H), 7.68 (d, J = 7.8 Hz, 1H), 7.57 (t, J = 7.7 Hz, 1H), 7.38 (d, J = 7.4 Hz, 1H), 7.34−7.24 (m, 1H), 7.14 (dq, J = 15.8, 7.7 Hz, 2H), 7.02 (dt, J = 15.6, 8.2 Hz, 1H), 6.93 (t, J = 6.7 Hz, 2H), 6.83 (d, J = 7.7 Hz, 1H), 5.54 (s, 1H), 4.03 (q, J = 7.1 Hz, 1H), 3.59 (dd, J = 23.0, 11.6 Hz, 2H), 2.51 (s, 2H), 1.67 (d, J = 8.1 Hz, 4H). ^13^C NMR (101 MHz, DMSO) δ 207.0, 204.2, 175.8, 170.9, 167.8, 163.9, 151.0, 147.1, 143.6, 139.8, 138.4, 130.3, 128.6, 126.9, 123.3, 121.2, 118.8, 113.6, 111.8, 110.4, 109.3, 106.2, 81.3, 60.4, 53.4, 51.9, 48.1, 45.6, 30.3, 26.0, 23.6, 21.2, 14.2. HR-MS (ESI): [M+H^+^] calcd for C_26_H_14_N_4_O_3_S: 462.0786 found: 462.1996.


**5**
**″**
**-bromo-2-iodo-2**
**″**
**-oxo-5**
**′**
**-sulfanylidene-1**
**″**
**,2**
**″**
**-dihydrodispiro[cyclohexane-1,2**
**′**
**-pyrrolidine-4**
**′**
**,3**
**″**
**-indole]-2,5-diene-3**
**′**
**,3**
**′**
**-dicarbonitrile (51).**


Brown solid (Yield: Met A 86%, Met B, 92%); m.p 130–132 °C. IR (NaCl) *v*(cm^−1^) 3323 (NH), 2214 (CN), and 1726 (CO). ^1^H NMR (400 MHz, CD_3_CN) δ 9.05 (s, 1H), 7.98 (d, J = 8.0 Hz, 1H), 7.66 (d, J = 9.8 Hz, 1H), 7.54 (t, J = 7.7 Hz, 1H), 7.12 (t, J = 7.8 Hz, 1H), 6.96 (d, J = 8.0 Hz, 1H), 6.91 (d, J = 8.5 Hz, 1H), 4.06 (q, J = 7.1 Hz, 1H), 2.28 (s, 6H), 1.24 (d, J = 14.9 Hz, 1H), 1.21 (t, J = 7.1 Hz, 1H). ^13^C NMR (101 MHz, CD_3_CN) δ 163.5, 149.7, 145.2, 140.3, 138.2, 128.6, 126.7, 124.1, 120.9, 115.6, 113.9, 112.6, 112.1, 111.3, 83.6, 60.7, 20.7, 14.0. HR-MS (ESI): [M+H^+^] calcd for C_18_H_10_BrIN_4_OS: 535.8803 found: 398.25.


**5″-bromo-4-methoxy-2″-oxo-5′-sulfanylidene-1″,2′′-dihydrodispiro[cyclohexane-1,2′-pyrrolidine-4′,3″-indole]-2,5-diene-3′,3′-dicarbonitrile (5j).**


Brown solid (Yield: Met A 91%, Met B, 83%); m.p 151–153 °C. IR (NaCl) *v*(cm^−1^) 3361 (NH), 2225 (CN), and 1710 (CO). ^1^H NMR (400 MHz, DMSO) δ 11.39 (s, 1H), 11.23 (s, 1H), 11.16 (d, J = 3.7 Hz, 1H), 7.91 (s, 1H), 7.76 (d, J = 8.5 Hz, 1H), 7.59 (d, J = 6.8 Hz, 1H), 7.14 (t, J = 7.7 Hz, 1H), 6.94 (t, J = 6.6 Hz, 1H), 5.58 (s, 1H), 4.09−3.94 (m, 1H), 2.09 (s, 8H), 1.99 (s, 1H), 1.23 (s, 1H), 1.18 (t, J = 7.1 Hz, 1H). ^13^C NMR (101 MHz, DMSO) δ 207.0, 163.9, 149.9, 146.1, 140.1, 128.2, 123.3, 120.9, 113.9, 113.2, 111.8, 82.7, 60.0, 31.3, 21.2, 14.5. HR-MS (ESI): [M+H^+^] calcd for C_19_H_13_BrN_4_O_2_S: 439.9942 found: 439.2704.


**{5″-bromo-3′,3′-dicyano-2″-oxo-5′-sulfanylidene-1″,2″-dihydrodispiro[cyclohexane-1,2′-pyrrolidine-4′,3″-indole]-2,5-dien-4-yl}azinic acid (5k).**


Brown solid (Yield: Met A 80%, Met B, 87%); m.p 157–159 °C. IR (NaCl) *v*(cm^−1^) 3345 (NH), 2220 (CN), and 1708 (CO). ^1^H NMR (400 MHz, DMSO) δ 11.39 (s, 1H), 11.23 (s, 1H), 10.63 (d, J = 28.5 Hz, 1H), 7.89 (s, 1H), 7.75 (d, J = 8.2 Hz, 1H), 7.57 (t, J = 7.8 Hz, 1H), 7.30 (d, J = 6.9 Hz, 1H), 7.22 (d, J = 8.3 Hz, 1H), 7.14 (d, J = 6.7 Hz, 1H), 6.92 (d, J = 8.4 Hz, 1H), 6.77 (d, J = 8.3 Hz, 1H), 4.03 (q, J = 7.1 Hz, 1H), 3.61 (d, J = 5.3 Hz, 1H), 3.48 (s, 1H), 2.09 (s, 3H), 1.99 (s, 1H), 1.70 (d, J = 13.9 Hz, 2H), 1.18 (t, J = 7.1 Hz, 1H). ^13^C NMR (101 MHz, DMSO) δ 207.0, 163.5, 149.9, 145.7, 140.1, 127.9, 120.9, 114.2, 112.9, 111.8, 82.4, 60.4, 54.1, 52.7, 30.9, 26.4, 20.8, 14.5. HR-MS (ESI): [M+H^+^] calcd for C_18_H_12_BrN_5_O_3_S: 457.9844 found: 457.7505.


**ethyl-5″-bromo-3′,3′-dicyano-2″-oxo-5′-sulfanylidene-1″,2″-dihydrodispiro[cyclohexane-1,2′-pyrrolidine-4′,3′-indole]-2,5-diene-2-carboxylate (5l).**


Brown solid (Yield: Met A 95%, Met B, 96%); m.p 150–152 °C. IR (NaCl) *v*(cm^−1^) 3358 (NH), 2220 (CN), and 1725 (CO). ^1^H NMR (400 MHz, DMSO) δ 11.11 (s, 1H), 10.56 (d, J = 28.2 Hz, 1H), 7.68 (s, 1H), 7.52 (s, 1H), 7.43 (d, J = 8.3 Hz, 1H), 7.24 (d, J = 8.3 Hz, 1H), 7.17 (d, J = 8.3 Hz, 1H), 7.09 (s, 1H), 6.85 (d, J = 8.3 Hz, 1H), 6.72 (d, J = 8.2 Hz, 1H), 5.69 (s, 1H), 5.51 (s, 1H), 3.96 (q, J = 7.1 Hz, 1H), 3.54 (s, 2H), 3.41 (s, 2H), 2.01 (s, 1H), 1.92 (s, 1H), 1.61 (s, 6H), 1.11 (t, J = 7.1 Hz, 1H). ^13^C NMR (101 MHz, DMSO) δ 204.2, 175.4, 170.8, 167.6, 164.6, 142.9, 138.7, 133.1, 130.1, 129.4, 128.4, 127.8, 126.9, 125.5, 125.2, 123.8, 121.7, 113.9, 112.5, 111.7, 110.2, 103.9, 60.2, 54.1, 52.7, 48.7, 30.0, 26.5, 23.4, 23.2, 14.5. HR-MS (ESI): [M+H^+^] calcd for C_21_H_15_BrN_4_O_3_S: 482.0048 found: 482.1327.

## 4. Conclusions

In this section, we show the facile synthesis of some new derivatives of spiro pyrrolidine compounds via a Michael condensation of 3-dicyanomethylene-2H-indol-2-ones (**3a**) or 5-bromo, 3-dicyanomethylene-2H-indol-2-ones (**3b**) with isothiocyanates derivatives under aqua and mechanochemical conditions. The described procedure is an appealing methodology for the Michael and Knoevenagel reaction due to the short reaction time, operational simplicity, high yield, and environmentally benign conditions. These spiro systems contain a range of functional groups, making them crucial building blocks for the diversity-oriented synthesis of spiro heterocyclic libraries with the potential to be employed as bioactive compounds.

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
