# Peer review of "Aqua/Mechanochemical Mediated Synthesis of Novel Spiro [Indole–Pyrrolidine] Derivatives"

_ijms, 2023, doi:10.3390/ijms24032307_

Round 1

Reviewer 1 Report

The manuscript ijms-2158103 is devoted to the design of novel spiro[indole-pyrrolidine] derivatives and can be interested to the specialists working in this field. The reviewed article is interesting and theme of the article meets the scope of the journal. Work is performed at sufficient scientific level; the results of investigation are professionally interpreted. However, it needs minor revision before publication. To improve the quality and perception of the manuscript I would suggest paying attention to following comments:

1) The abstract contains general phrases. It needs to be rewritten. The abstract should reflect the main results presented in the article.

2) Experimental part must be carefully checked and corrected. Given the specifics of the journal, it is necessary to add a full experimental chemical part and provide physicochemical (yields, melting points, etc.) and spectral data for key compounds. Data for less important derivatives can be left in the Supplementary information.

3) Malononitrile is written incorrectly in reaction schemes (Scheme 2, line 133; Scheme 4, lline 166; Figure 5, line 180). It is necessary to replace H2C-CN2 with NCCH2CN or CH2(CN)2.

4)
There are some grammar and orthographical errors in the manuscript, which should be corrected.

My decision is m
inor revision.

Author Response

1) The abstract contains general phrases. It needs to be rewritten. The abstract should reflect the main results presented in the article.

Thank you for the observation. This has been rewritten as suggested.

2) Experimental part must be carefully checked and corrected. Given the specifics of the journal, it is necessary to add a full experimental chemical part and provide physicochemical (yields, melting points, etc.) and spectral data for key compounds. Data for less important derivatives can be left in the Supplementary information.

Thank you for the valuable suggestion. The experimental details have been added to the main manuscript as you rightly suggested

3) Malononitrile is written incorrectly in reaction schemes (Scheme 2, line 133; Scheme 4, lline 166; Figure 5, line 180). It is necessary to replace H2C-CN2 with NCCH2CN or CH2(CN)2.

Thank you. This error has been corrected.
4) There are some grammar and orthographical errors in the manuscript, which should be corrected.

Thank you. We have thoroughly checked and revised the manuscript

Reviewer 2 Report

The present manuscript, titled „Aqua/Mechanochemical Mediated Synthesis of Novel Spiro [Indole-pyrrolidine] Derivatives“ is focused on the present problem of finding new possibilities in the synthesis of small molecules.
Spiro compounds are very interesting and may have potential medical applications. Current works deal with both natural pseurotin and its synthetic analogues. It has been shown in in vitro and in vivo models that they may have medicinal potential.
If you want, you can use the latest articles with other interesting natural spiro compounds, in the introduction:
Vasicek et al. 2020: Natural pseurotins and analogs thereof inhibit activation of B-cells and differentiation into the plasma cells; Phytomedicine
Mosejova et al. 2021: Pseurotin d induces apoptosis through targeting redox sensitive pathways in human lymphoid leukemia cells; Antioxidants
Vasicek et al. 2020: Natural pseurotins inhibit proliferation and inflammatory responses through the inactivation of STAT signaling pathways in macrophages; Food and chemical Toxicoligy

The chosen „Communication“ manuscript type is standardly structured, legibly written without major typographical errors.
I have only a few comments on the manuscript:
- carefully read through the text and unify the names of the compounds (for example line 160 „product 5A ..“ vs. 5a in all text.
- I would put section 4. Conclusions right after 2. Results and Dscussion and lastly I would put Materials and Methods section. The reader would seamlessly follow on from the Results and Discussion from the Conclusions. 

Author Response

Vasicek et al. 2020: Natural pseurotins and analogs thereof inhibit activation of B-cells and differentiation into the plasma cells; Phytomedicine
Mosejova et al. 2021: Pseurotin d induces apoptosis through targeting redox sensitive pathways in human lymphoid leukemia cells; Antioxidants
Vasicek et al. 2020: Natural pseurotins inhibit proliferation and inflammatory responses through the inactivation of STAT signaling pathways in macrophages; Food and chemical Toxicoligy

Thank you so much. We have updated the manuscript with the reference above.

 carefully read through the text and unify the names of the compounds (for example line 160 „product 5A ..“ vs. 5a in all text.

Thank you for the observation. This error has been corrected.
- I would put section 4. Conclusions right after 2. Results and Dscussion and lastly I would put Materials and Methods section. The reader would seamlessly follow on from the Results and Discussion from the Conclusions. 

Thank you for the suggestion. This has been done accordingly.